# Pediatric growth hormone treatment in Italy: A systematic review of epidemiology, quality of life, treatment adherence, and economic impact

**Massimiliano Orso** [1]*, **Barbara Polistena** [1,2], **Simona Granato** [3], **Giuseppe Novelli** [4], **Roberto Di Virgilio** [4], **Daria La Torre** [5], **Daniela d'Angela** [1,2], **Federico Spandonaro** [1,6]

1 C.R.E.A. Sanità (Centre for Applied Economic Research in Healthcare), Rome, Italy, 2 University of Rome Tor Vergata, Rome, Italy, 3 Medical Department, Pfizer Italia, Rome, Italy, 4 Health Economics & Outcomes Research, Pfizer Italia, Rome, Italy, 5 Global Medical Affairs, Pfizer Rare Disease, Rome, Italy, 6 San Raffaele University, Rome, Italy

* massi.orso@hotmail.it

## Abstract

### Objectives

This systematic review aims to describe 1) the epidemiology of the diseases indicated for treatment with growth hormone (GH) in Italy; 2) the adherence to the GH treatment in Italy and factors associated with non-adherence; 3) the economic impact of GH treatment in Italy; 4) the quality of life of patients treated with GH and their caregivers in Italy.

### Methods

Systematic literature searches were performed in PubMed, Embase and Web of Science from January 2010 to March 2021. Literature selection process, data extraction and quality assessment were performed by two independent reviewers. Study protocol has been registered in PROSPERO (CRD42021240455).

### Results

We included 25 studies in the qualitative synthesis. The estimated prevalence of growth hormone deficiency (GHD) was 1/4,000–10,000 in the general population of children; the prevalence of Short Stature HOmeoboX Containing gene deficiency (SHOX-D) was 1/1,000–2,000 in the general population of children; the birth prevalence of Turner syndrome was 1/2,500; the birth prevalence of Prader-Willi syndrome (PWS) was 1/15,000. Treatment adherence was suboptimal, with a range of non-adherent patients of 10–30%. The main reasons for suboptimal adherence were forgetfulness, being away from home, pain/discomfort caused by the injection. Economic studies reported a total cost for a complete multi-year course of GH treatment of almost 100,000 euros. A study showed that drug wastage can amount up to 15% of consumption, and that in some Italian regions there could be a considerable over- or under-prescribing. In general, patients and caregivers considered the GH

**Data Availability Statement:** All relevant data are within the manuscript and its Supporting Information files.

**Funding:** This study was sponsored by Pfizer. SG, GN, RDV and DLT are employees of Pfizer; they contributed in writing and reviewing the manuscript.

**Competing interests:** Simona Granato, Giuseppe Novelli, Roberto Di Virgilio and Daria La Torre are employees of Pfizer. Barbara Polistena declares to have received in the last 5 years payments or honoraria for lectures, presentations, speakers bureaus, manuscript writing or educational events from the following commercial sources: Allergan, Amgen, Astellas, Baxter, BMS, Boehringer-Ingelheim, Celgene, Eli Lilly, Janssen Cilag, Jazzpharma, Mylan, Nestlé HS, Novartis, Novo Nordisk, Pfizer, Roche, Sanofi, Servier, Shire, Takeda, Teva; in addition, she received consulting fees from UCB. Federico Spandonaro declares to have received in the last 5 years payments or honoraria for lectures, presentations, speakers bureaus, manuscript writing or educational events from the following commercial sources: Allergan, Amgen, Astellas, Baxter, BMS, Boehringer-Ingelheim, Celgene, Eli Lilly, Janssen Cilag, Jazzpharma, Mylan, Nestlé HS, Novartis, Novo Nordisk, Pfizer, Roche, Sanofi, Servier, Shire, Takeda, Teva; in addition, he received consulting fees from Amgen. All other authors declare that they have no competing interests. This does not alter our adherence to PLOS ONE policies on sharing data and materials.

treatment acceptable. There was a general satisfaction among patients with regard to social and school life and GH treatment outcomes, while there was a certain level of intolerance to GH treatment among adolescents. Studies on PWS patients and their caregivers showed a lower quality of life compared to the general population, and that social stigma persists.

## Conclusion

Growth failure conditions with approved GH treatment in Italy constitute a significant burden of disease in clinical, social, and economic terms. GH treatment is generally considered acceptable by patients and caregivers. The total cost of the GH treatment is considerable; there are margins for improving efficiency, by increasing adherence, reducing drug wastage and promoting prescriptive appropriateness.

## Introduction

Growth hormone (GH), also known as somatropin, is recommended as the main treatment option for growth failure in children with one of the following conditions: growth hormone deficiency (GHD), Turner syndrome (TS), growth retardation in infants born small for gestational age (SGA), Prader-Willi syndrome (PWS), growth retardation due to chronic kidney failure (CKF), and growth retardation associated with a defect in the Short Stature HOmeoboX Containing (SHOX) gene. This treatment has also been shown to be effective in improving body composition, muscle mass and strength, exercise capacity, glucose and lipid profile, bone metabolism, and quality of life in adults with GHD diagnosed in adulthood or childhood.

Although there is a wide international literature on somatropin, focused on clinical effectiveness, safety and economic aspects, we were interested in verifying the available evidence specifically for Italy. Our aim was to identify studies conducted in Italy concerning GH therapy in non-adult subjects, through a systematic review of the literature. To our knowledge, there is not a published systematic review on this topic available.

Specifically, we were interested in estimating the epidemiology of GH-treated growth failure conditions in Italy, verifying treatment adherence, assessing the economic impact of GH treatment and describing the quality of life of patients treated with GH and/or caregivers.

## Methods

This systematic review is based on a study protocol registered in PROSPERO (CRD42021240455). The reporting of the review follows the PRISMA reporting checklist (S1 Checklist).

### Review questions

The research questions that this systematic review aims to address, are:

1. to assess the prevalence and the incidence in Italy of pediatric conditions approved as indication for GH treatment: GHD, TS, CKF, PWS, SGA, SHOX gene deficiency;

2. to assess adherence to GH treatment in Italy and identify factors that could facilitate or hamper the adherence;

3. to assess the economic impact of GH treatment in Italy;

4. to assess the quality of life of patients treated with GH and their caregivers in Italy.

## Searches

Systematic searches have been performed in PubMed, Embase and Web of Science, looking for papers published from January 2010 to March 2021, describing national or international studies. Only articles written in English or Italian were included. The complete search strategies are reported in S1 Table.

## Inclusion criteria

**Participants/population.** Non-adult subjects (infants, children and adolescents) with an indication for the treatment with GH. All the following conditions were included: GHD, TS, CKF, PWS, SGA, SHOX gene deficiency.

**Intervention(s), exposure(s).** Growth hormone treatment.

**Comparator(s)/control.** All the comparators (placebo, other treatments, no intervention) were considered.

## Types of study to be included

- Research question 1 (epidemiology): epidemiological studies assessing the prevalence/incidence of the conditions with an indication for the treatment with GH in pediatrics;

- Research question 2 (adherence): studies of any design assessing adherence to GH treatment;

- Research question 3 (economic impact): economic studies assessing the economic impact of the treatment with GH, in terms of total costs, cost-effectiveness, cost-consequence, cost-utility, drug wastage costs, and other relevant economic measures.

- Research question 4 (quality of life): studies of any design assessing the quality of life of patients treated with GH and their caregivers.

For all the research questions, only studies performed in Italy, or international multicenter studies with at least one Italian center, were included.

## Main outcomes

- Research question 1 (epidemiology): prevalence and incidence of conditions treated with GH in Italy; in absence of Italian estimates, worldwide estimates were considered;

- Research question 2 (adherence): adherence to GH treatment;

- Research question 3 (economic impact): cost-effectiveness and other economic analyses of GH treatment;

- Research question 4 (quality of life): quality of life of patients treated with GH and their caregivers.

## Measures of effect

- Research question 1 (epidemiology): prevalence and incidence;

- Research question 2 (adherence): medication possession ratio (MPR), proportion of days covered (PDC), proportion of doses administered vs prescribed, percentage of adherent patients;

 

- Research question 3 (economic impact): total costs, drug wastage costs, incremental cost-effectiveness ratio (ICER), quality-adjusted life year (QALY), etc.;

- Research question 4 (quality of life): results reported in qualitative studies (questionnaires, interviews, etc.) aimed to assess the quality of life in patients treated with GH and their caregivers.

## Literature screening process

The study selection process was performed by two independent reviewers. Any disagreement was solved through discussion and, when necessary, a third reviewer was contacted. Study selection was conducted in two phases. Initially, the reviewers assessed the records through the titles and abstracts screening, according to the predefined inclusion criteria. In the second phase, the reviewers evaluated the full-text of the potential eligible studies. The final studies included in the review were described in the main text and in the tables, while a list of excluded studies along with the reasons for exclusion have been published as (S2 Table). Bibliographic references were managed using the EndNote X7.4 software.

## Data extraction

Data extraction was performed by one reviewer and verified by another reviewer using a standardized form. The following information was extracted from the included studies: bibliographic data (first author, publication year and title), study characteristics (study design, location, study period, number of centers, sample size), participant characteristics (disease/condition, gender, age), treatment characteristics (type of GH, dosage and duration), comparator characteristics (type of comparator, dosage and duration), study outcomes (incidence/prevalence, adherence measures, economic measures, quality of life scores).

Specifically, for each research question we extracted the following information.

Research question 1 (epidemiology)—secondary studies: number of included studies and their references, disease, epidemiological estimates (incidence, prevalence, etc.); primary studies: disease, study population (N), epidemiological estimates. None of the primary studies identified in this review are in common with those included in the secondary studies.

Research question 2 (adherence)—definition of non-adherence / non-adherent patients (%), reasons for non-adherence, factors influencing (barriers / facilitators) the adherence.

Research question 3 (economic impact)—type of economic analysis, type of costs / discount rate / economic perspective / reference year, results.

Research question 4 (quality of life)—methods used for assessing QoL, subjects interviewed, results.

## Quality assessment of included studies

The quality assessment was performed by one reviewer and verified by another reviewer, using appropriate checklists according to the study design of included studies. In particular, the following checklists were used: for prevalence studies, the checklist by Hoy et al. [1]; for cohort studies, the CASP checklist [2]; for qualitative studies, the CASP checklist [3]; for case series, the Institute of Health Economics (IHE) Quality Appraisal Checklist for Case Series Studies [4]; economic studies were evaluated by the Consensus Health Economic Criteria (CHEC) list [5].

## Strategy for data synthesis

According to study protocol, a quantitative synthesis of the results through a meta-analysis would have been performed only in the case three or more (included) studies reported on the

same outcome for any research question; moreover, to be combined, the studies would have to be similar in terms of PICO (patient/population, intervention, comparator, outcome). Assuming a high level of heterogeneity between the included studies, it was planned to perform random effects meta-analyses, providing cumulative estimates along with 95% confidence intervals. The heterogeneity between the included studies would have been assessed using the $I^2$ statistic, considering a statistical significance level of p <0.05 for all analyses. The STATA 13/SE software was indicated for the statistical analyses. It was also specified in the protocol that, in all the cases where a meta-analysis was not considered feasible, the results would have been presented narratively.

## Results

The literature search performed in Medline (via PubMed), Embase and Web of Science identified 1,383 records overall. After removing duplicates, 924 records were screened through title/abstract and 50 of these were considered eligible for full-text evaluation. Twenty-five articles were excluded after full-text screening, while 25 articles [6–30] were included in the final analysis. A list of excluded studies along with reasons for exclusion is provided in S2 Table. Among the 25 included studies, 9 studies [7, 13, 14, 17, 19, 25, 28–30] were used to address the research question 1 (epidemiology), 10 studies [6, 8–12, 18, 20, 22, 23] the research question 2 (adherence), 5 studies [15, 16, 21, 26, 28] the research question 3 (economic impact), and 4 studies [8, 21, 24, 27] the research question 4 (quality of life); three studies [8, 21, 28] contributed for more than one research question. The literature selection process is depicted in the Fig 1 (PRISMA Flow Diagram).

The results are presented narratively, without attempting a quantitative synthesis (meta-analysis) as the studies included in this review were heterogeneous in terms of study design, objectives and PICO (patient/population, intervention, comparator, outcome).

### Epidemiology

To address the research question 1 (epidemiology), we identified 9 studies [7, 13, 14, 17, 19, 25, 28–30], 5 of which were secondary studies [14, 25, 28–30] and 4 were primary studies [7, 13, 17, 19] reporting epidemiological data about the considered diseases. The main characteristics of the included studies are described in the Table 1 (secondary studies) and Table 2 (primary studies).

Among the secondary studies, we included 3 narrative reviews [14, 25, 29], 1 economic analysis with a narrative review on epidemiology [28], and 1 commentary [30].

The 2 reviews by Cicognani [14] and Nicolosi [25] reported the prevalence of SHOX deficiency in ISS children. The review by Cicognani [14] included 7 studies [31–37], 6 of which reported international estimates [31–34, 36, 37], and 1 reported Italian estimates [35]. Most of the included studies were in common between the two reviews. Cicognani [14] reported a prevalence range of 1.1%-14%, with an expected prevalence in children of 1/2,000, while Nicolosi [25] reported a prevalence range of 1.1%-15%, with an expected prevalence in children of 1/1,000.

Other two included studies were those of Tornese [29, 30], related to SGA. The first article [29] was a narrative review about SGA, in which it is stated that, by definition, SGA children should be 2% of the population (– 2 SDS of birth weight/length, corresponding to the 2nd percentile). This hypothetical prevalence is lower than the prevalence reported in the three population studies included in the review: 3.1% (Finland) [41] (SGA was defined as– 2 SDS of birth weight compared to a reference population), 5.5% (Sweden) [42], and 3.5% (Japan) [40] (in the last two studies, SGA was defined as – 2 SDS of birth weight or birth length compared to a

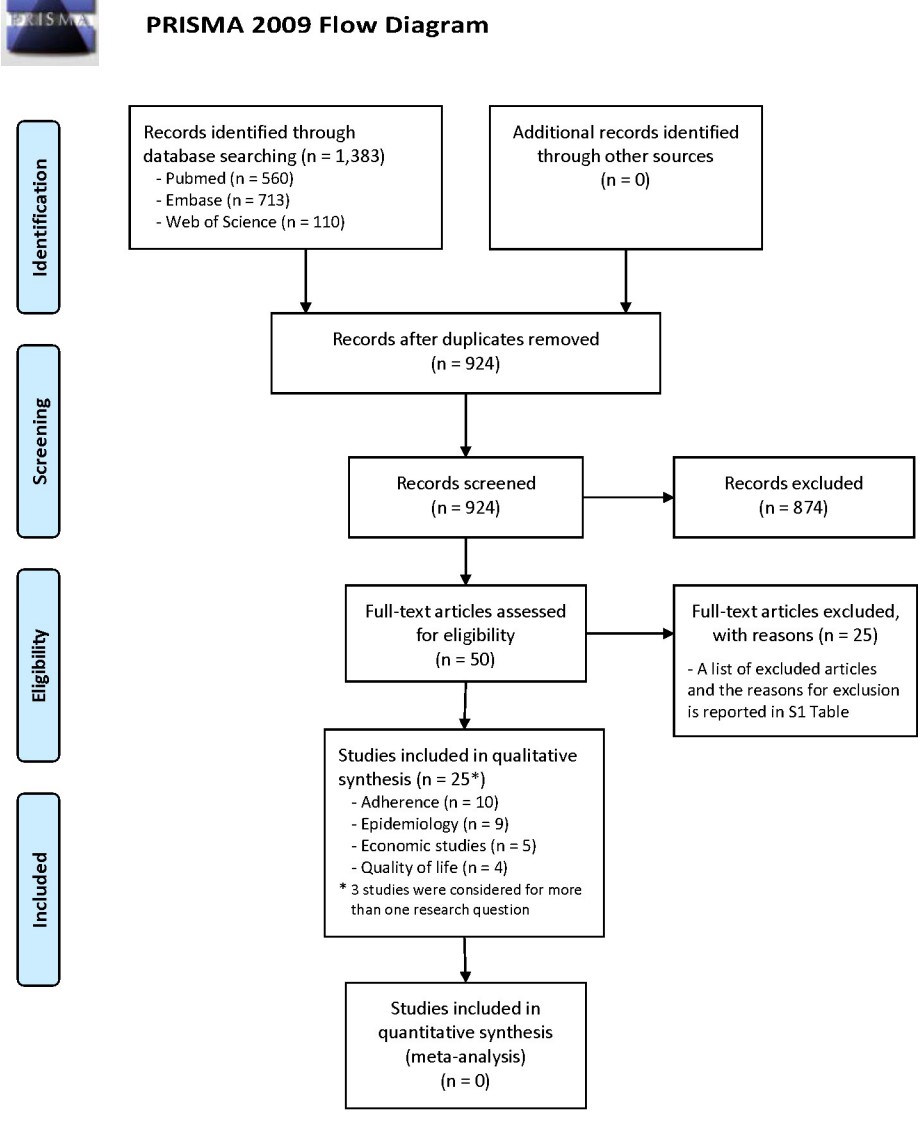

**Fig 1. PRISMA 2009 flow diagram.**

reference population). In the population study conducted in Sweden, Karlberg et al. [42] reported that 12% of the children with SGA had growth retardation at 2 years of age. According to this ratio, considering a minimum prevalence of 2% of SGA in the general population, the hypothetical prevalence of short stature in SGA children would be 0.24% (1: 417) at two years of age. A Japanese study [40] verified the prevalence of short-stature children born SGA by studying a cohort of nearly 30,000 children born over a three-year period and re-evaluated at 3 years of age: the prevalence was 0.06% (1: 1,800).

The other study by Tornese et al. [30] was a commentary based on data from the Italian National Registry of Growth Hormone therapy (RNAOC), in the years 2011–2017. This study showed that the prevalence of SGA patients treated in 2017 was 0.37/100,000 (1.79/100,000

**Table 1. Characteristics of included secondary studies for epidemiology.**

| Study ID | Study design | Included studies, N (Setting) | Ref. included studies | Disease | Epidemiological estimates |
|---|---|---|---|---|---|
| Cicognani 2010 [14] | Narrative review | n = 7 (6 international and 1 Italian) | • Rao 1997 [31] | SHOX defect (SHOX-D) | • Prevalence of SHOX gene mutations in idiopathic short stature: range between included studies: 1.1%-14%. |
| | | | • Binder 2000 [32] | | • Estimated population prevalence of SHOX-D in children: 1/2,000. |
| | | | • Rappold 2002 [33] | | |
| | | | • Binder 2003 [34] | | |
| | | | • Stuppia 2003 [35] | | |
| | | | • Huber 2006 [36] | | |
| | | | • Jorge 2007 [37] | | |
| Nicolosi 2010 [25] | Narrative review | n = 8 (7 international and 1 Italian) | • Rao 1997 [31] | SHOX-D | • Overall prevalence of SHOX-D in idiopathic short stature (ISS): range 1.1%-15.0%. |
| | | | • Binder 2000 [32] | | • Expected population prevalence of SHOX-D in children: 1/1,000. |
| | | | • Rappold 2002 [33] | | |
| | | | • Stuppia 2003 [35] | | |
| | | | • Huber 2006 [36] | | |
| | | | • Jorge 2007 [37] | | |
| | | | • Rappold 2007 [38] | | |
| | | | • Chen 2009 [39] | | |
| Tornese 2019a [29] | Narrative review | n = 3 (international) | • Fujita 2016 [40] | • SGA children | • By definition, SGA should be 2% of the population (− 2 SDS correspond to the 2nd percentile) |
| | | | • Raisanen 2013 [41] | • GHD | • SGA with growth failure at 2 years of age: hypothetical prevalence 0.24% (1/417). |
| | | | • Karlberg 1995 [42] | | • SGA prevalence reported in included studies: 3.1% (Finland) [41], 5.5% (Sweden) [42], 3.5% (Japan) [40] |
| | | | | | • Prevalence of children (3-year-old) eligible for treatment as SGA: 1/1,800 (Japan) [40] |
| | | | | | • GHD, estimated prevalence in children: 1/4,000-1/10,000 (source: not stated). |
| Tornese 2019b [30] | Commentary based on registry data | N.A. | N.A. | SGA children | • The prevalence of SGA patients treated in 2017 was 0.37/100,000 (1.79/100,000 in children 0–15 years). |
| | | | | | • Incidence of treated SGA patients: 0.42/100,000 per year (mean of the years 2011–2017). |
| | | | | | Data source: National Registry of Growth Hormone therapy (RNAOC), years 2011–2017. |
| Spandonaro 2014 [28] | Economic analysis with a narrative review on epidemiology | n = 1 (Italian) | Migliaretti 2006 [43] | GHD | Index of exposure to treatment in Piedmont, Italy (Source: Piedmont GH registry, years 2002–2004): 9.44 subjects per 10,000 residents <18 years |
| | | n = 1 (International) | Sybert & McCauley 2004 [44] | TS | Birth prevalence: 1/2,500 |
| | | n = 2 (International) | • Lindgren 1999 [45] | PWS | Birth prevalence: 1/15,000 (average between 1/10,000—Lindgren 1999 [45] and 1/29,000—Butler 1990 [46]) |
| | | | • Butler 1990 [46] | | |
| | | n = 1 (Italian) | Ardissino 2003 [47] | CKF | Birth prevalence: 0.001% |
| | | n = 1 (International) | Karlberg 1995 [42] | SGA | Birth prevalence: 5.4% |

**Table 2. Characteristics of included primary studies for epidemiology.**

| Study ID | Study design | Setting / Study period | Disease, Study population (N) | Epidemiological estimates |
|---|---|---|---|---|
| Behnisch 2019 [7] | Design: observational, prospective, multicenter study | Setting: S. Orsola-Malpighi Hospital Bologna (Italy) | Disease: Chronic kidney disease (CKD) | Prevalence of growth failure between CKD children: 11/46 (23.9%) |
| | | Enrolment period: Jan 2010 – May 2012; Follow-up: until Dec 2018 | Study population: CKD children (n = 46) | |
| Chiavaroli 2016 [13] | Design: retrospective review of obstetric and delivery records | Setting: Chieti province (Italy) | Disease: SGA* | Incidence in the year 1993: 8.3% |
| | | Period: 1993–2013 | *defined on a birthweight and/or length <10th percentile (<−1.28 SD) | Incidence in the year 2013: 7.6% |
| | | | Study population: n = 5,896 | |
| Genoni 2018 [17] | Design: prospective cohort study | Setting: Novara (Italy) | Disease: SHOX-D | Prevalence of SHOX-D in a population of short-statured children: 6,8% |
| | | Period: 2012–2015 | Study population: 281 children with short stature (height <3rd percentile) | |
| Kodra 2019 [19] | Design: registry study | Setting: Italy | Disese: PWS | The annual incidence of PWS was 0.08 and 0.10 per 100 000 in data from NRDR and from HDD, respectively. |
| | | Period: 2012–2014 | Study population: 65,889 cases of rare diseases; 143 cases of PWS | |

considering children 0–15 years). The incidence of treated SGA patients was 0.42/100,000 per year (mean of the years 2011–2017). The authors commented this data as extremely lower than expected, concluding that many short children born SGA are still not properly identified in Italy, and therefore not treated with GH, or misdiagnosed as GHD.

Another secondary study included was the review paper by Spandonaro et al. [28], aimed to determine how much of the variability in GH consumption in Italy is actually due to differences in clinical practice, and how much to waste. This literature review identified 6 articles [42–47] reporting epidemiological data on 5 GH-treated conditions; the epidemiological estimates are showed in Table 1.

In addition, our literature search retrieved 4 primary epidemiological studies [7, 13, 17, 19] (Table 2). Behnisch et al. [7] analyzed a cohort of 594 patients with CKD from 12 European countries; the prevalence of growth failure varied between countries from 7 to 44%, while the prevalence in Italy was 23.9%.

Chiavaroli et al. [13] described a retrospective review of obstetric records over a 20-year period in the Chieti province (Italy). The incidence of SGA children was stable in the study period, ranging from 8.3% in 1993 to 7.6% in 2013. Prevalence in this study is higher compared to studies described above, due to a different definition of SGA (birthweight and/or length <10th percentile, i.e. <−1.28 SD).

The paper by Genoni et al. [17] described a prospective cohort study conducted in Novara (Italy) in the period 2012–2015. This study found a prevalence of SHOX-D of 6.8% in a population of 281 children with short stature.

In another epidemiological study included, Kodra et al. [19] presented data on PWS from 2 Italian registry, the Italian National Rare Diseases Registry (NRDR) and the National Hospital Discharge Database (HDD). The annual incidence of PWS in a 3-year period (2012–2014) was 0.08 and 0.10 per 100,000 in data from NRDR and from HDD, respectively.

In summary, in the included studies:

- the estimated prevalence of GHD in the pediatric population (international estimates) is about 1/4,000–10,000, while the prevalence of GHD in Italy was indirectly estimated through the index of exposure to GH treatment in the Piedmont region (Italy), that is 9.44 subjects per 10,000 residents <18 years;

- the minimum hypothetical prevalence of SGA children is 2%, with a range of 3.1–5.5%; the hypothetical prevalence of SGA children with growth retardation at 2 years is 0.24%, while in a Japanese study it was 0.06%;

- the prevalence of SHOX-D in children with idiopathic short stature ranges between 1.1–15.0%, while prevalence in the general population of children is about 1/1,000-1/2,000 (international estimates);

- the birth prevalence of Turner syndrome is 1/2,500 (international estimates);

- the birth prevalence of PWS is about 1/15,000 (international estimates), while the annual incidence in Italy is reported between 0.08 and 0.10 per 100,000 in registry studies;

- the birth prevalence of CKD in the general population is 0.001%, and the prevalence of growth failure among CKD children in Italy is 23.9%.

**Quality assessment.** The methodological quality of secondary included studies was not assessed because we assumed that it was low, based on study design (narrative review or commentary). The quality of primary studies was assessed through the Hoy checklist [1], a tool designed to assess the risk of bias in population-based prevalence studies. Three studies [7, 13, 17] fulfilled 7/10 criteria for a low risk, with an item of the checklist considered not applicable (likelihood of non-response bias). In these three studies, the external validity has been considered limited, due to the small sample size and the risk of selection bias in the study sample. One study [19] was judged to have a low risk of bias (9/10 items satisfied, and one not applicable).

## Adherence

To address research question 2 (adherence), we identified 10 studies [6, 8–12, 18, 20, 22, 23], 5 of which report on adherence to GH treatment in general, using different approaches to assess adherence [6, 9, 10, 18, 23], and 5 studies describing the adherence with the Easypod™ device (Saizen®, Merck Serono S.A.—Geneva, Switzerland) [8, 11, 12, 20, 22], an electronic auto-injector that automatically records the patient's adherence to treatment. The characteristics of included studies are reported in Table 3.

The multicenter study by Bagnasco [6] was conducted in 46 Italian pediatric centers, with the aim to evaluate self-reported adherence to GH therapy in a representative sample of Italian children and adolescents, and to identify the determinants of poor adherence. Non-adherent patients, defined as those that missed ≥1 injection over a typical week during the last 12 months of GH treatment, were 24.4%. The main reasons reported for non-adherence are: being away from home (33.3%), forgetfulness (24.7%), not feeling well (12.9%), and pain (10.3%). Among the barriers to adherence, that is, factors associated with low adherence, low level of parent education and longer treatment duration are reported. Among the facilitators, that is, factors associated with high adherence, convenience of the injection device and awareness of the consequences of non-adherence are reported.

The international, multicenter study by Bozzola et al. [8] was conducted in 206 centers in 15 countries, with the aim to assess the opinions of users of the Easypod™ auto-injector and the adherence to the treatment. Non-adherent patients, defined as those that missed more than two daily injections per month or six daily injections in a 3-month period, were 12.5% of the entire study population (n = 824; GHD 66.4%, TS 9.8%, CRF 1.7%, SGA 15.3%, other 6.8%), of which 10.3% were naïve, and 18.3% experienced users. With regard to the Italian centers involved, non-adherent patients were 20.9% (15.4% naïve vs 34.4% experienced). Among

**Table 3. Characteristics of included studies for adherence.**

| Study ID | Study design | Setting / Study period | Population | Definition of non-adherence / non-adherent patients (%) | Reasons for non-adherence | Adherence barriers / facilitators |
|---|---|---|---|---|---|---|
| Bagnasco 2017 [6] | Survey | Setting: 46 Italian pediatric centers<br><br>Period: November 2015 –May 2016 | Subjects (n = 1,007) aged 6–16 years, of both sexes, on GH treatment for at least 6 months<br><br>Diagnosis: not reported. | ≥1 injection missed over a typical week during the last 12 months of GH treatment<br><br>Non-adherent patients: 24.4% | Being away from home (33.3%), forgetfulness (24.7%), not feeling well (12.9%), and pain (10.3%). | Barriers: low level of parent education, longer duration of treatment, need to convince the child to inject, and low level of awareness of the consequences of not following treatment.<br><br>Facilitators: convenience and satisfaction with the device, overall satisfaction with the treatment |
| Bozzola 2011 [8] | Survey | Setting: 206 centers, across 15 countries<br><br>Enrolment period: 1.5 years; Survey period: 3 months. | Subjects (Overall: n = 824; Italian: n = 112), median age 11 years (range 1–18 years); sex: M 56%.<br><br>Diagnosis: GHD (66.4%), TS (9.8%), CRF (1.7%), SGA (15.3%), other (6.8%). | More than two daily injections missed per month or six daily injections missed in the 3-month period<br><br>Non-adherent patients: Overall population 12.5%, (10.3% naïve, 18.3% experienced); Italian patients: 20.9% (15.4% naïve, 34.4% experienced). | Forgetfulness (43.7%), device not working (18.2%), running out of cartridges/needles (12.9%), being away from home (12.6%) | Not reported |
| Bozzola 2014 [9] | Case series | Setting: not reported<br><br>Period: not reported | Subjects: GHD children (n = 106), mean age 10.5 ± 3.5 years; sex: 73 M, 33 F. | Different non-adherence levels (from missing occasional doses per week to discontinuing the therapy).<br><br>Growth failure was observed in 11/106 children treated with GH after a period of good growth response to long-term GH therapy; among them, 10/11 admitted poor adherence. | Complex treatment regimen, pain. | Not reported |
| Buzi 2016 [10] | Narrative review | Setting and period: 12 international studies included, published between 1993 and 2011 | Subjects treated with GH enrolled in the included studies (range: n = 17–6,487)<br><br>Diagnosis: not stated (10 studies); CRF (2 studies). | Different definitions of non-adherence across the included studies.<br><br>Non-adherent patients: range 5–82%. | Not reported | Barriers: discomfort, complex treatment regimens, age, personal factors, understanding of the benefits of treatment.<br><br>Facilitators: use of automatic injection devices or increasingly fine needles; use of needle-free devices. |
| Cardinale 2019 [11] | Observational, retrospective | Setting: 6 Italian centers<br><br>Period: January 2015 –September 2015 | Subjects (n = 90), mean age 11.9±3.4 years; sex: 52 M, 38 F.<br><br>Diagnosis: GHD (92%), SGA (6%), TS (2%). | Ratio between actual days of treatment and planned days of treatment. Patients were classified according to different adherence levels: ≤50%, 50–60%, 60–70%, 70–80%, 80–90%.<br><br>Mean adherence: 70% ± 13% (SD), corresponding to 649 actual days of treatment on 977 planned days. | Not reported | Not reported |

*(Continued)*

**Table 3.** (Continued)

| Study ID | Study design | Setting / Study period | Population | Definition of non-adherence / non-adherent patients (%) | Reasons for non-adherence | Adherence barriers / facilitators |
|----------|--------------|------------------------|------------|----------------------------------------------------------|---------------------------|-----------------------------------|
| Centonze 2019 [12] | Observational, prospective, multicenter | Setting: 22 Italian centers | Subjects: GHD children (n = 73) naïve to GH treatment, mean age 9.8 (SD 3.2) years; sex: M 38, F 35. | Adherence definition: % of injections administered vs prescribed. | Not reported | Not reported |
| | | Period: study duration was 3 years | | Mean adherence: >85% (1st year: 88.5%; 2nd year: 86.6%; 3rd year 85.7%) | | |
| Giavoli 2020 [18] | Survey | Setting: single center in Milan, Italy. | Subjects: GHD children (n = 107), mean age 11.3 (SD 3.5) years; | Pediatric population: adherence was measured by the Morisky Medication Adherence Scale (8 points = high; 6.0–7.9 points = medium; <6.0 points = low). | Children: problems related to drug supply; motivational problems. Adults/Transition age: problems related to drug supply. | Not reported |
| | | Period: April 2020 | Adults (n = 92), mean age 54 (SD 12) years; | | | |
| | | | Transition age (n = 9), mean age 19 (SD 2) years | Adults/Transition age: patients who declared taking > 80% of the total number of the prescribed GH injections were considered adherent. | | |
| | | | | Children: high adherence in 82% of patients. | | |
| | | | | Adults/Transition age: 94% of subjects reported an acceptable adherence. | | |
| Loche 2016 [20] | Observational, prospective | Setting: 10 Italian centers | Subjects: GHD children (n = 79), median age 10 years (IQR 9–12): sex: 52 M, 27 F. | Adherence: 92% of injections administered vs prescribed. | Not reported | Not reported |
| | | Period: March 2010 –January 2013 | | Adherence data available for 53/79 children. 30/53 reported ≥300 injections during the 12 months of observation. | | |
| | | | | 17/30 (56.7%) had an adherence ≥ 92%. | | |
| Maggio 2018 [22] | Observational, retrospective | Setting: single center in Palermo, Italy. | Subjects: children (n = 40), mean age 11.2 (SD 2.3); sex: 27 M, 13 F. | Adherence: % of injections administered vs prescribed. | Not reported | Barriers: adherence was inversely related to patients' age, duration (years) and frequency (n. of doses/ week) of treatment |
| | | Period: 2009–2016 | Diagnosis: GHD (65%), SGA (22.5%), TS (12.5%). | Mean treatment adherence: 92.2%. | | |
| Marcianò 2018 [23] | Observational, retrospective, based on healthcare databases | Setting: six Italian centers | Subjects: n = 4,493 GH naïve users, median age 12 years (IQR 9–21); sex: males/females ratio = 1.3 | 2,428 up to 4,493 (54.0%) patients discontinued the therapy for at least 60 days | Not reported | Barriers: persistence inversely related to patients' age |
| | | Period: 2009–2014 | Diagnosis (available for 2,430 children): GHD (88.8%), CRF (4.4%), TS (2.3%), PWS (1.5%), SGA (1.3%). | | | |

the reasons for poor adherence, the following have been reported: forgetfulness (43.7%), device malfunction (18.2%), running out of cartridges/needles (12.9%), being away from home (12.6%).

Another study by Bozzola et al. [9] aimed to examine adherence to GH therapy in GHD patients in whom a growth deficit has been observed after a period of good response to therapy. In addition, new strategies are proposed for the management of non-adherence to long-term GH therapy. Among 11/106 children in whom growth failure was observed, 10 admitted poor adherence. Among patients, there were different levels of non-adherence, from missing occasional doses per week to discontinuing the therapy. Reported reasons for non-adherence included complex treatment regimen and pain.

Buzi et al. [10] carried out a narrative review of international studies investigating adherence to GH treatment. The authors reported that the definitions of non-adherence and the methods used for its assessment vary considerably between the included studies. Non-adherent patients ranged from 5% to 82% in the different countries. Among the barriers to adherence, the following were reported: discomfort, complex therapeutic regimens, age, personal factors, level of understanding the benefits of treatment. Among the facilitators: use of automatic injection devices or increasingly fine needles, use of needle-free devices.

The paper by Cardinale et al. [11] describes a multicenter study conducted in 6 Italian centers, with the aim of evaluating adherence to GH treatment in patients with growth deficiency, monitored using the Easypod ™ device. Adherence was defined as the ratio between actual days of treatment and planned days of treatment. Patients were classified according to different adherence levels: ≤50%, 50–60%, 60–70%, 70–80%, 80–90%. Non-adherent patients were on average 30% of the total.

Centonze et al. [12] conducted a multicenter study in 22 Italian centers. The study reported the 3-year prospective adherence data of the Italian cohort of naïve GHD children extrapolated from the Easypod Connect Observational Study (ECOS) database. Adherence is defined as the percentage of injections recorded by the device compared to those prescribed. The mean adherence was > 85% (1st year: 88.5%; 2nd year: 86.6%; 3rd year 85.7%).

The paper by Giavoli [18] described the results of a survey carried out during the period of the COVID-19 emergency among patients treated with GH and their families. The objective of the survey was to collect information on the clinical conditions of the patients, on their laboratory results, as well as reinforce a recommendation to stay at home as indicated in the emergency phase. In addition, possible changes in adherence to therapies during the pandemic period were verified. In the pediatric population, adherence was measured by the Morisky Medication Adherence Scale (MMAS); in this scale, a total score of 8 indicates a high level of adherence, a score between 6.0 and 7.9 a medium level of adherence, and a score less than 6 a low level of adherence. The adherence of adults/transition age patients was self-reported: patients who declared taking > 80% of the total number of the prescribed GH injections were considered adherent.

In the pediatric GHD population, 82% of patients showed high adherence, while in the adult/transition population 94% declared an acceptable adherence. The reasons for poor adherence in the pediatric population were related to drug supply problems and motivational problems, while in the adult/transition population mainly drug supply problems.

The paper by Loche et al. [20] described a multicenter study conducted in 10 Italian centers, with the aim of evaluating adherence in GHD patients treated with GH using Easypod™. Adherence was defined as the proportion of injections correctly administered during the observational period out of the expected total number of injections; patients were considered fully adherent if they had an adherence rate ≥ 92%. Adherence data was available for 53/79 children: 30/53 reported ≥300 injections in the 12 months of the study; 17/30 (56.7%) had an adherence rate of ≥ 92%.

Maggio et al. [22] conducted a single-center, observational, retrospective study based on real-world data, with the aim of describing the correlation between the effectiveness of GH

treatment administered via Easypod ™ and adherence. The average adherence (% of injections administered vs prescribed) in the children involved was 92.2%. Adherence was inversely correlated with the patient's age, the duration of therapy, the number of GH doses. In the study, height gain did not reach a significant correlation with treatment adherence.

The paper by Marcianò [23] reported a drug-utilization study using administrative databases. The aim of the study was to explore the pattern of use of biosimilar and originator GH in six Italian centers. The study showed that 2,428 of 4,493 naïve patients (54.0%) discontinued therapy for at least 60 days over the 6-year study period. There were no statistically significant differences in persistence to treatment between biosimilar GH and originator. Most of those who discontinued were intermittent users (39.3%), that is, they restarted GH therapy after at least 60 days of interruption; in their paper, Marcianò et al. [23] have speculated that treatment discontinuation could be due to patients' reduced compliance or lack of perceived benefit from GH therapy. Another possible reason for discontinuing GH therapy may be the occurrence of adverse reactions; in fact, more than 25% of patients who discontinued therapy had a new diagnosis of diabetes or cancer or has been hospitalized at least once after discontinuation. Another reason for discontinuation could be reaching final height (7.9% discontinued at age 15–17). The remainder of the subjects who discontinued therapy (27.7%) were thought to be probably related to the lack or loss of GH efficacy or to patients' lack of compliance.

**Quality assessment.** The methodological quality of 5 included studies [11, 12, 20, 22, 23] was assessed through the CASP Cohort checklist. The range of items fulfilling criteria for a low risk was 8-10/12. There were some concerns about the identification of all the possible confounding factors, the length and completeness of follow-up, and the generalizability of the results.

The methodological quality of 3 included studies [6, 8, 18] was assessed through the CASP Qualitative checklist. The 3 articles satisfied 9/10 items; no information was found to address the item related to whether the relationship between researcher and participants has been adequately considered.

The study by Bozzola et al. (2014) [9] was assessed through the IHE checklist for case series. This paper satisfied 13/20 items of the checklist, not providing sufficient information about the study design and the characteristics of the study population.

The quality of the narrative review by Buzi et al. [10] was not assessed and considered low due to the study design.

## Economic impact

To address the research question 3 (economic impact), we identified 5 studies [15, 16, 21, 26, 28]. The characteristics of included studies are reported in the Table 4. One of these studies [28] has already been considered for the research question about epidemiology, while another study [21] has been also considered for the research question about quality of life.

The paper by Drube et al. [15] described an international consensus statement on GH treatment in children with chronic kidney disease and includes a cost-effectiveness analysis based on the median price of one gram of GH in eight European countries, including Italy. This study showed the total drug-related costs for four different scenarios: 1) 5-year-old patient at the start of treatment and 2-year treatment duration: total cost € 13,000; 2) 5-year-old patient at the start of treatment and 5-year treatment duration: total cost € 37,900; 3) 12-year-old patient at the start of treatment and 2 years of treatment: total cost € 27,100; 4) 12-year-old patient at the start of treatment and duration of treatment of 5 years: total cost € 80,100. The corresponding incremental cost per centimeter gained in height as an adult for the 4 scenarios was, respectively, € 1,800, € 5,300, € 3,800 and € 11,100.

**Table 4. Characteristics of included studies for economic impact.**

| Study ID | Study design / Setting / Study period | Disease / Population (N) | Type of economic analysis | Type of costs / Discount rate / Perspective / Reference year | Results |
|---|---|---|---|---|---|
| Drube 2019 [15] | Design: consensus statement including a cost-effectiveness analysis<br><br>Setting: 8 European countries (including Italy)<br><br>Period: 2018 | Chronic kidney disease<br><br>Population: not applicable | Cost-effectiveness analysis | • total drug-related costs* | • The total drug-related costs for a patient aged 5 or 12 years at start of treatment range from €13,000 to €37,900 and €27,100 to €80,100, respectively, depending on the length of treatment (2 or 5 years). |
| | | | | • incremental cost per centimeter gained* | • The corresponding incremental cost per centimeter gained at adult height for a patient aged 5 or 12 years at the start of treatment ranges from €1,800 to €5,300 and from €3,800 to €11,100, respectively, depending on the length of treatment (2 or 5 years). |
| | | | | * based on a cost of €22 per 1 mg of GH (median cost between 8 European countries) | |
| Foo 2019 [16] | Economic evaluation<br><br>Setting: Italy<br><br>Period: not described | GHD<br><br>Population: 10,000 hypothetical patient profiles | Cost-consequence analysis | • total costs of GH treatment<br>• drug wastage costs<br>• incremental cost per centimeter gained | • Compared to other drugs, somatropin had the second highest total cost for a complete multi-year GH treatment including wastage (€ 96,710) but had the second lowest cost per cm gained (€ 7,699 / cm). |
| | | | | Discount rate: 3%. | • In the scenario analysis, somatropin with Easypod had the lowest cost per cm gained (€4,708/cm) amongst all of the compared treatments (Saizen®, NutropinAq®, Humatrope®, Genotropin®, Omnitrope®, Norditropin SimpleXx®, Zomacton®) |
| Lopez-Bastida 2016 [21] | Cross-sectional study (survey)<br><br>Setting: 8 European countries (including Italy)<br><br>Period: September 2011—April 2013 | PWS<br><br>Population: 261 patients (175 were <18 years); 48 patients in Italy (32 were <18 years) | Prevalence-based cost-of-illness analysis | • total costs | • The average annual costs* ranged from € 3,937 to € 70,083 between countries (Italy: € 33,787); the reference year for unit prices was 2012. |
| | | | | • direct healthcare costs | • Direct healthcare costs* ranged from € 458 to € 17,695 (Italy: € 4,974), direct non-healthcare costs ranged from € 1,387 to € 52,389 (Italy: € 28,813). |
| | | | | • direct non-healthcare costs<br><br>Perspective: society<br>Reference year: 2012 | *per patients <18 years. |
| Pagani 2011 [26] | Cohort study with a cost-effectiveness analysis<br><br>Setting: Pavia, Italy<br>Period: not described | GHD<br>Population: 12 GHD vs 12 bioinactive GH children treated with GH | Cost-effectiveness analysis | • total costs<br>• ICER | • There were no significant differences in cost/height gain between GHD (€ 1,925±653) and bioinactive GH children (€ 1,640±631). |
| | | | | | • There were also no significant differences in cost/year of therapy between GHD (€ 12,348±2,018) and bioinactive GH children (€ 11,355 ±1,748). |
| Spandonaro 2014 [28] | Economic evaluation and healthcare utilization analysis<br>Setting: Italy<br>Period: 2012 | Conditions with an indication to GH treatment<br>Population: 11,329 subjects treated with GH | Model estimating the prevalence of patients with indication for GH treatment and waste estimation in the Italian regions | Drug consumption in terms of mg. per capita | The study showed over-prescription and potential under-prescription, ranging from 20% to 40% less than estimated theoretical consumption to over 200% more. Wastage, at the level of a single device, could amount to up to 15% of consumption. |

In the study by Foo et al. [16], patients who received somatropin with Easypod ™ gained, on average, 3.2 cm more than patients who received other GH treatments. In the base case analysis, which considered the list prices of drugs, somatropin administered with the Easypod ™ device (Saizen®) was the second in terms of average total cost for a complete multi-year GH treatment, including the cost deriving from drug wastage (Omnitrope® € 57,024, NutropinAq ® € 73,304, Humatrope® € 78,098, Norditropin SimpleXx® € 90,689, Genotropin® € 96,188, Saizen® € 96,710, Zomacton® € 104,895), but it was the second least expensive per cm gained (Omnitrope® € 6,058/cm, Saizen® € 7,699/cm vs NutropinAq® € 7,787/cm, Humatrope® € 8,297/cm, Norditropin SimpleXx® € 9,634/cm, Genotropin® € 10,218/cm, Zomacton® € 11,143/cm). Considering published tender prices (scenario analysis), Saizen® had the lowest cost per cm gained (Saizen® € 4,708/cm, Omnitrope® € 5,065/cm, NutropinAq® € 6,179/cm, Humatrope® € 6,442/cm, Genotropin® € 6,528/cm, Norditropin SimpleXx® € 6,917/cm, Zomacton® € 9,851/cm) across all compared treatments.

The paper by Lopez-Bastida et al. [21] aimed to determine the economic burden from a societal perspective and the health-related quality of life (HRQOL) of patients with PWS in Europe. A survey was conducted on 261 patients (48 in Italy) to estimate the social/economic costs including three dimensions of costs: direct healthcare costs, direct non-healthcare costs (formal and informal care) and labour productivity losses. The study results per patients < 18 years (175 in total, 32 in Italy) showed that the average annual costs ranged from € 3,937 to € 70,083 between countries (€ 33,787 in Italy). Direct healthcare costs ranged from € 458 to € 16,695 (€ 4,974 in Italy), and direct non-healthcare costs ranged from € 1,387 to € 52,389 (€ 28,813 in Italy).

Pagani et al. conducted a study [26] including two groups of subjects: 12 children without GH deficiency (bioactive GH) and 12 children with GH deficiency; both groups were exposed to GH treatment. The study found a significant ($p < 0.05$) increase in growth rate in both groups during the first year of GH treatment (non-GHD: –1.7 to 5.4 standard deviation score (SDS); GHD: –1.5 to 4.7 SDS). There was no statistically significant variation between the two groups in the difference between final height and target height. Similarly, no significant difference in cost/height gain was found between GHD children (€ 1,925 ± 653) and children with bioactive GH (€ 1,640 ± 631). Furthermore, no significant differences were found in the annual cost of therapy between GHD (€ 12,348 ± 2,018) and children with bioactive GH (€ 11,355 ± 1,748).

The study by Spandonaro et al. [28] was conducted to determine how much variability in GH consumption in Italy is actually due to differences in clinical practice and how much waste is related to the devices used for drug administration. The model estimated that, for some devices, waste can amount to up to 15% of consumption. The consumption model, applied to the Italian population, has shown that in various Italian regions one can assume either an excessive prescription or a potential under-prescription, ranging from 20 to 40% less to over 200% more than the estimated theoretical consumption.

**Quality assessment.** The quality of included studies was evaluated through the CHEC list, a 19-item checklist. The quality of Spandonaro et al. [28] was not assessed, because the items of the CHEC list were not considered applicable to this study. The quality of the other four studies was as follows: Drube et al. [15] 13/19 "yes", Foo [16] 16/19, Pagani [26] 12/19, while Lopez-Bastida [21] obtained 12/19 "yes", with 7 items retained not applicable because related to cost-consequence / cost-effectiveness analyses. Among the checklist items judged as not fulfilled, there were: two studies [16, 26] did not adopt a societal perspective; two studies [15, 26] did not specify the sources of valuation for each cost price and their reference year; two studies [15, 26] did not convert all costs to one single year, based on a motivated discount rate, and did not perform sensitivity analyses; three studies [15, 16, 26] did not discuss about

generalizability of the results and on ethical and distributional issues; two studies [15, 26] did not include statements about conflict of interest.

## Quality of life

To address the research question 4 (quality of life), we identified 4 studies [8, 21, 24, 27]. Another study [48] describing the validation of the Quality of Life in Short Stature Youth (QoLISSY) questionnaire in Italy was selected but not included in the systematic review since it does not report quality of life results; this study is briefly described in the Discussion section. The characteristics of included studies are reported in Table 5. One of these studies [8] has already been considered for the research question about adherence, while another study [21] has been also considered for the research question about economic impact.

The two studies by Lopez-Bastida [21] and Ragusa [27] focus mainly on the disease (PWS) rather than on the GH treatment. We decided to include them in our analysis because we were interested in analyzing the quality of life in patients with PWS, which is one of the diseases for which GH treatment is approved in Italy, and we assumed that most patients were treated. In the Lopez-Bastida paper [21], information on medications used by PWS patients was obtained

Table 5. Characteristics of included studies for quality of life.

| Study ID | Study design / Setting / Study period | Disease / Population (N) | Methods used for assessing QoL | Subjects interviewed | Results |
|---|---|---|---|---|---|
| Bozzola 2011 [8] | Survey<br><br>Setting: 206 centers, across 15 countries<br><br>Enrolment period: 1.5 years; Survey period: 3 months | Diseases: GHD, TS, CKF, SGA, other<br><br>Subjects (Overall: n = 824; Italian: n = 112), median age 11 years (range 1–18 years); sex: M 56%. | Questionnaire | Children or parents | • Most children liked the auto-injector Easypod™: over 80% gave the top two responses from five options for ease of use (720/779), speed (684/805) and comfort (716/804). |
| | | | | | • 38.5% (300/780) of children reported pain on injection, but over half of children (210/363) considered the pain to be less or much less than expected. |
| | | | | | • 91.8% (732/797) of children/parents would continue using the device. |
| Lopez-Bastida 2016 [21] | Cross-sectional study (survey)<br><br>Setting: 8 European countries (including Italy)<br><br>Period: September 2011—April 2013 | PWS<br><br>Population: 261 patients (175 <18 years of age); 48 patients in Italy (32 <18 years of age) | EuroQol 5-domain (EQ-5D) questionnaire | Patients and caregivers | PWS patients*:<br><br>• the mean EQ-5D index score ranged between 0.40 and 0.81 (Italy: 0.40); |
| | | | | | • the mean EQ-5D visual analogue scale score ranged between 51.25 and 62.63 (Italy: 56.15). |
| | | | | | *Quality of life measures have been reported only for adult patients. |
| | | | | | Caregivers: |
| | | | | | • the mean EQ-5D index score ranged from 0.73 to 0.82 (Italy: 0.82); |
| | | | | | • the mean EQ-5D visual analogue scale ranged from 70.26 to 81.52 (Italy: 77.81). |
| Marini 2016 [24] | Quantitative/ qualitative study using narrative medicine<br><br>Setting: 11 Italian centers<br><br>Period: April 2013 – December 2013 | GHD<br><br>Population: 182 narratives (67 patients; 72 parents; 7 siblings; healthcare professionals: 19 stories + 17 parallel charts) | Collection of narratives | Patients, parents, siblings, healthcare professionals | The study showed recurrent signals of intolerance among adolescents and the worry of not being well informed about side effects among parents. |
| | | | | | Both the GHD patients and their parents appreciated the work of healthcare professionals and were satisfied for the outcomes of therapy. |

(Continued)

**Table 5.** (Continued)

| Study ID | Study design / Setting / Study period | Disease / Population (N) | Methods used for assessing QoL | Subjects interviewed | Results |
|---|---|---|---|---|---|
| Ragusa 2020 [27] | Qualitative study<br><br>Setting: 10 Italian centers<br><br>Period: October 2018 –July 2019 | PWS<br><br>Population: 21 children and 34 adults with PWS and 138 caregivers | Collection of narratives | Patients and caregivers | • Diagnosis and current management of PWS: disbelief, displacement, anger and pain represented the most recurrent emotions expressed by caregivers; food-seeking behaviours emerged as the most challenging event within the domestic context. PWS patients were aware of the importance of following a diet. |
| | | | | | • Living with PWS in relationships and in social contexts: most of caregivers reported fatigue, chaos, all-encompassing assistance and using tested routines to better manage food-seeking behaviours. Caregivers have attempted to maintain their hobbies, while relationships external to the family are difficult to preserve. |
| | | | | | • Work and future perspectives: most of caregivers had to change their job after the birth of their child with PWS; adult participants with PWS demonstrated self-realisation through work. |
| | | | | | In general, narratives showed that PWS management affects relationships and work-life balance and that social stigma persists. |

NFPA, nonfunctioning pituitary adenoma.

from questionnaires and their costs calculated. In the Ragusa study [27] there were references to GH treatment in interviews with various patients and caregivers.

In the international multicenter study of Bozzola et al. (2011) [8] which included 206 centers in 15 countries, including Italy, the Easypod ™ device was found to be acceptable by users. The device was described by 89.1% (716/804) of the participants as convenient/very comfortable to use, and 91.8% (732/797) of respondents said they would like to continue using the device.

The paper by Lopez-Bastida [21], previously described in this review, showed that in adult PWS patients the mean EuroQol 5-domain (EQ-5D) index score (where 0 corresponds to death and 1 corresponds to perfect health), ranged between 0.40 and 0.81 (Italy: 0.40), while the mean EQ-5D visual analogue scale score (where 0 corresponds to worst imaginable health state and 100 to best imaginable health state) ranged between 51.25 and 62.63 (Italy: 56.15). Although this study included patients with a mean age of 14 years (range 6–22), quality of life measures have been reported only for adult patients. Among caregivers, the mean EQ-5D index score ranged from 0.73 to 0.82 (Italy: 0.82), and the mean EQ-5D visual analogue scale ranged from 70.26 to 81.52 (Italy: 77.81).

The qualitative study by Marini et al. [24], conducted in 11 Italian centers on patients with GHD and their caregivers highlighted a general satisfaction among patients with regard to social and school life and GH treatment outcomes, while there was a certain level of intolerance to GH treatment among adolescents. One-third of the adolescents said about the GH treatment "I don't like it but I have to do it"; 39% of them lived the medical visits with boredom and impatience, and 50% of them suffered for the daily task. Most of parents (67%) had concern of not being well informed about side effects. Both the GHD patients and their parents

appreciated the work of healthcare professionals and were satisfied with the outcomes of therapy. The narratives of healthcare professionals showed the need to find a more engaging way of communication with adolescents, and to reassure the families about therapy and its possible effects, in order to increase the adherence to the therapy.

The multicenter study by Ragusa [27] et al., which involved 10 Italian centers, investigated the impact of PWS on patients and caregivers through narrative medicine, to understand the problems related to daily life, the needs and resources of PWS patients and their caregivers and provide insights into clinical practice. Disbelief, displacement, anger and pain represented the most recurrent emotions expressed by caregivers in an attempt to adapt to the situation and its criticalities. From the caregivers' narratives, food-seeking behaviors emerged as the most challenging event to manage in the home context. Relations outside the family were difficult to preserve, imposing a radical change in social life. Sixty-two percent of family caregivers had to change job after the birth of a child with PWS.

**Quality assessment.** Two studies have been assessed through the CASP Qualitative checklist: Marini [24] and Ragusa [27] showed a complete reporting according to the checklist (10/10 items). The quality assessment of the last two studies [8, 21] has been reported in the previous paragraphs.

## Discussion

With regard to the epidemiology of diseases/conditions with an indication for GH treatment, we found mostly international estimates, with prevalence estimate for GHD in children ranging from 1/4,000 to 1/10,000 [29]; this estimate is similar but with a wider range than that reported in a HTA report by the NICE [49] (1/3,500–4,000). An Italian study [30] based on the GH registry of the Piedmont Region indirectly estimated the prevalence of GHD through the index of exposure to GH treatment, that is 9.44 subjects per 10,000 residents <18 years. It should be noted that the GHD prevalence estimated by the Piedmont GH registry is higher than the prevalence reported in the literature; however, it should be considered as an indirect estimate which may differ from the real prevalence in the general population. The authors of the original study from which this estimate was reported [43], in the discussion state that their prevalence estimates are higher of those reported in previous studies, probably due to different patient selection criteria, the cut-off used for tests allowing the diagnosis of partial or complete GHD, and the improvements of diagnostic techniques (higher sensitivity).

Turner syndrome occurs in about 1/2,500 girls [44] (international estimates). Prader-Willi syndrome occurs in 1/15,000 live births [45, 46] (international studies), while the annual incidence in Italy is 0.08–0.10 per 100,000 [19]. Regarding PWS, the international estimates are only apparently different from the Italian ones. In fact, the PWS prevalence of 1/15,000 (about 7 per 100,000) reported in the literature is a birth prevalence, while the reported annual incidence in Italy refers to the entire Italian population (which was on average 60.2 million per year in the period 2012–2014). Estimating the birth prevalence from the annual incidence on the total population, it would be: 37 new PWS cases/year (annual average of PWS patients in the age group 0–10 years) out of 527,845 live births (annual average of live births over the years 2012–2014), that is 7 per 100,000, exactly the same birth prevalence reported in the literature.

Short stature secondary to chronic kidney failure affects 7–44% of patients with CKF in 12 European countries, while in Italy it affects 23.9% of patients [7]. The birth prevalence of SGA children varies between 3.1% and 5.5% [40–42] (estimates from international studies); the hypothetical prevalence of SGA children with growth retardation at 2 years is 0.24%, while in a Japanese population-based study it was 0.06% [29]. The prevalence range of SHOX-D in

children with idiopathic short stature is 1.1–15.0%, while prevalence in the general population of children is about 1/1,000-1/2,000 (international estimates) [14, 25]. This wide range could be due to different sample size of included studies, patient selection criteria, ethnicity, and different methods used for the genetic analysis. The quality of the included studies can be considered acceptable, with some concerns about the generalizability of the results; the main limitation of included studies is that they mostly provide international estimates and therefore there is a lack of evidence related to Italian studies. Further Italian studies are required to have a better comprehension of the burden of disease.

As regards adherence among patients treated with GH, the studies identified are very heterogeneous in terms of study objectives, methods, and definitions of non-adherence adopted; their methodological quality was judged moderate/good.

In the included studies, non-adherence in Italy varies between 10% and 25%, values consistent with international estimates, although a narrative review [10] reported a very wide range of non-adherence of 5–82%. In the time frame considered by the systematic review, various contributions were found relating to the Easypod™ device which, depending on the studies, report a non-adherence rate of 12.5–30%.

Among factors linked to decreased adherence identified in the included studies, the most cited are: forgetfulness, being away from home, pain/discomfort caused by the injection, complex therapeutic regimen, device malfunction, low level of parental education. On the contrary, the factors cited as favoring greater adherence are: convenience/usability of the device for drug administration, awareness of the consequences related to non-adherence, use of automatic injection devices or of increasingly fine needles, use of needle-free devices. Non-adherence to GH treatment, in addition to being associated with reduced therapeutic efficacy, can also lead to increased economic costs. In fact, in the case of undetected non-adherence, a patient may be subjected to further diagnostic tests to re-evaluate potential false diagnoses. Furthermore, in case of non-adherent GHD patients, their GH doses may be increased, with an augmented risk for treatment-related adverse events and increased costs [10, 50].

Regarding the economic impact of GH therapy we included 5 studies, with a variable methodological quality (from 12 to 16 out of 19 items of the checklist fulfilled). An international included study [15] showed that the median total costs (in 8 European countries, for which a median cost of € 22 per mg was estimated) of GH treatment, for a patient aged 5 or 12 at the start of treatment, range from € 13,000 to € 37,900 and from € 27,100 to € 80,100, respectively, depending on the duration of the treatment (2 or 5 years); the incremental cost per centimeter gained in adult height for a 5 or 12 year old patient at the start of treatment ranges from € 1,800 to € 5,300 and from € 3,800 to € 11,100, respectively, depending on the duration of treatment.

According to a comparative study [16], the somatropin treatment with EasypodTM (Saizen®) was more effective than the other treatments, allowing to reach 3.2 cm extra height. Despite a higher total cost than almost all other drugs, Saizen® has the second lowest cost per cm earned (€ 7.699/cm earned). This cost is in line with that reported by NICE [49], which estimated the cost of GH therapy to be around £ 6,000 per cm of final height for patients with GHD, from £ 15,800 to £ 17,300 per cm for Turner syndrome, £ 7,400 to £ 24,100 per cm for CRI and approximately £ 7,030 per cm for PWS.

In a report from the Canadian Agency for Drugs and Technologies in Health (CADTH) [51], the economic evaluation of somatropin treatment in children with Turner syndrome estimated an ICER of Canadian dollar (CAD) 23,630 for cm of height gained and CAD 243,078 per QALY gained. A US report on the economic evaluation of somatropin treatment in children with GHD estimated an ICER of $ 36,995 per QALY for the 5–16 years old cohort and $ 42,556 per QALY gained for the cohort aged between 3 and 18 years [49].

An Italian study [28] showed that product waste can reach up to 15% of consumption for some GH treatment devices; moreover, that in the Italian regions the prescriptive levels of GH treatment come to be from 20 to 40% less, up to over 200% more, compared to the estimated theoretical consumption.

Finally, regarding the quality of life of patients treated with GH, the included studies show that the treatment is generally considered acceptable by patients and their caregivers. A multi-center study that also includes Italy [8] showed that the Easypod™ device was found to be acceptable by patients who were satisfied with its use; the only negative point concerns the pain experienced with the injection (38.5% of the answers). Another study [24] showed satisfaction with social and school life, and with the results of treatment with GH, in the majority of Italian patients with GHD and caregivers; instead, there is a certain level of intolerance to GH treatment among adolescents (in terms of annoyance, boredom and impatience with respect to medical visits and the long treatment duration), and parental concern of not being well informed about side effects of the treatment. The included studies were considered having a low risk of bias.

In addition to the studies included for quality of life, we found the study by Quitmann [48] et al. that carried out the translation, cultural adaptation, and validation of the Quality of Life in Short Stature Youth (QoLISSY) questionnaire in Italy, and the results revealed comparability of the contents with the data of the five European countries in which it was originally validated. The psychometric quality of the Italian version of QoLISSY is considered satisfactory by the authors and therefore the tool is considered ready to be used in Italian patients and their parents.

This systematic review of the literature, specifically aimed to find studies on the Italian population, highlighted that the evidence produced on national data is scarce. For both the epidemiological and economic aspects, the evidence for Italy is largely obtained through the adaptation of international data or data from Italian centers included in international multi-center studies. Some more information can be found on adherence and regarding the quality of life of patients and caregivers.

Probably due to the time frame used for the systematic review (2010–2021), many of the included studies are related to a specific injector, for which evidence were produced in various studies on Italian data.

## Strengths and limitations

The main strengths of this review are: it is based on a study protocol registered in PROSPERO; the reporting of the review follows the PRISMA guidelines; a comprehensive literature search was performed in three electronic databases; the literature selection process, data extraction and quality assessment were performed independently by two reviewers.

Potential limitations regard the time frame considered for the literature search (2010–2021) and that we considered only published studies, without searching the grey literature; these limits could have reduced the sensitivity of the search.

## Conclusions

In conclusion, pediatric conditions with short stature with an indication for GH treatment constitute a significant burden of disease, both in clinical, social, and economic terms. Epidemiological estimates were heterogeneous and mostly based on international studies. Further studies are needed that report incidence and prevalence of these conditions in Italy.

Quality of life studies show that, as expected, growth failure in children and adolescents can be associated with social stigma and levels of quality of life lower than subjects with normal height. However, patients and caregivers consider GH treatment as acceptable.

The total costs of a complete multi-year treatment with GH reach almost € 100,000 per patient. Even in the presence of studies that evaluate the cost-effectiveness of treatment with GH, margins for achieving more efficiency remain, as shown by some analyses on consumption and therefore on prescriptive appropriateness, and also treatment adherence should be improved.

## Supporting information

**S1 Checklist. PRISMA-2009 reporting checklist.**
(DOC)

**S1 Table. Literature search strategies.**
(DOCX)

**S2 Table. Excluded studies.**
(DOCX)

## Author Contributions

**Conceptualization:** Massimiliano Orso, Barbara Polistena, Daniela d'Angela, Federico Spandonaro.

**Data curation:** Massimiliano Orso.

**Formal analysis:** Massimiliano Orso, Barbara Polistena, Federico Spandonaro.

**Funding acquisition:** Simona Granato.

**Investigation:** Massimiliano Orso, Barbara Polistena.

**Methodology:** Massimiliano Orso, Barbara Polistena, Federico Spandonaro.

**Project administration:** Barbara Polistena, Simona Granato, Daniela d'Angela, Federico Spandonaro.

**Supervision:** Barbara Polistena, Federico Spandonaro.

**Writing – original draft:** Massimiliano Orso, Barbara Polistena, Simona Granato, Giuseppe Novelli, Roberto Di Virgilio, Daria La Torre, Daniela d'Angela, Federico Spandonaro.

**Writing – review & editing:** Massimiliano Orso, Barbara Polistena, Simona Granato, Giuseppe Novelli, Roberto Di Virgilio, Daria La Torre, Daniela d'Angela, Federico Spandonaro.

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
