## [Decision Letter · Decision Letter 0]

7 Dec 2021

PONE-D-21-34417Pediatric growth hormone treatment in Italy: a systematic review of epidemiology, quality of life, treatment adherence, and economic impactPLOS ONE

Dear Dr. Orso,

Thank you for submitting your manuscript to PLOS ONE. After careful consideration, we feel that it has merit but does not fully meet PLOS ONE’s publication criteria as it currently stands. Therefore, we invite you to submit a revised version of the manuscript that addresses the points raised during the review process.

We look forward to receiving your revised manuscript.

Kind regards,

Mohamed A Yassin, MD

Academic Editor

PLOS ONE

Journal Requirements:

"I have read the journal's policy and the authors of this manuscript have the following competing interests: 

Simona Granato, Giuseppe Novelli, Roberto Di Virgilio and Daria La Torre are employees of Pfizer.

Barbara Polistena declares to have received in the last 5 years payments or honoraria for lectures, presentations, speakers bureaus, manuscript writing or educational events from Allergan, Amgen, Astellas, Baxter, BMS, Boehringer-Ingelheim, Celgene, Eli Lilly, Janssen Cilag, Jazzpharma, Mylan, Nestlé HS, Novartis, Novo Nordisk, Pfizer, Roche, Sanofi, Servier, Shire, Takeda, Teva; in addition, she received consulting fees from UCB.

Federico Spandonaro declares to have received in the last 5 years payments or honoraria for lectures, presentations, speakers bureaus, manuscript writing or educational events from Allergan, Amgen, Astellas, Baxter, BMS, Boehringer-Ingelheim, Celgene, Eli Lilly, Janssen Cilag, Jazzpharma, Mylan, Nestlé HS, Novartis, Novo Nordisk, Pfizer, Roche, Sanofi, Servier, Shire, Takeda, Teva; in addition, he received consulting fees from Amgen.

All other authors declare that they have no competing interests."

We note that you received funding from a commercial source: "Allergan, Amgen, Astellas, Baxter, BMS, Boehringer-Ingelheim, Celgene, Eli Lilly, Janssen Cilag, Jazzpharma, Mylan, Nestlé HS, Novartis, Novo Nordisk, Pfizer, Roche, Sanofi, Servier, Shire, Takeda, Teva, and UCB."

Please include your amended Competing Interests Statement within your cover letter. We will change the online submission form on your behalf."

Additional Editor Comments:

needs major revision in order to be considered for publication

Reviewers' comments:

Reviewer's Responses to Questions

**Comments to the Author**

1. Is the manuscript technically sound, and do the data support the conclusions?

Reviewer #1: Yes

Reviewer #2: Yes

2. Has the statistical analysis been performed appropriately and rigorously? 

Reviewer #1: N/A

Reviewer #2: Yes

3. Have the authors made all data underlying the findings in their manuscript fully available?

Reviewer #1: Yes

Reviewer #2: Yes

4. Is the manuscript presented in an intelligible fashion and written in standard English?

Reviewer #1: Yes

Reviewer #2: Yes

5. Review Comments to the Author

Reviewer #1: The manuscript is a systematic review of the epidemiology of disorders requiring growth hormone supplementation and specific other growth hormone treatment related issues. The authors focused on studies including Italian national data. The overview is well presented with detailed methodology and informative tables containing results. The discussion is mostly summarizing the results with a few major conclusions. These are particularly exposing the possibilities to decreasing costs of GH therapy, which is of public importance in light of very high treatment costs. In general, I have no major comments regarding the manuscript. Despite the fact that part of the authors are employed in pharmaceutical companies producing GH therapeutics, I did not notice any biased reporting.

Reviewer #2: Summary: This is a systematic review that addresses 4 major questions regarding GH use in children in Italy: the prevalence and incidence of pediatric conditions approved as indication for GH treatment: GHD, TS, CKF, PWS, SGA, SHOX gene deficiency, adherence to GH treatment, the economic impact of GH treatment, and the quality of life of patients treated with GH and their caregivers.

Major comments:

1. Abstract, Results section: In the paragraph discussing prevalence, state the population, i.e., general population, referred patients, etc.

2. Page 7, Data extraction: Clarify that there was no overlap between primary and secondary studies.

3. Page 13, 2nd paragraph: Add a discussion about the different definitions of SGA used in the studies.

4. Page 14, 1st summary bullet: Does the general population mentioned in the prevalence of GHD include both children and adults? There is a striking difference between the prevalence noted in the international studies and that in Italy. (see below)

5. The majority of the epidemiological studies are international and most have only 1 study in Italy resulting in a wide prevalence range.

6. Page 20, 1st paragraph: Include the various definitions of non-adherence in the text.

7. Page 22, 2nd paragraph: Include the reasons for discontinuation in the paper by Marciano. Was this by choice or due to prescription/supply issues?

8. Discussion: Explain the differences between international and Italian estimates of GHD and PWS prevalence.

9. Page 31: the quality of life assessment in patients with PWS does not seem to be specifically regarding GH treatment. Please clarify.

Minor comments:

1. Page 34, last paragraph: Expand “CAD.”

6. PLOS authors have the option to publish the peer review history of their article (what does this mean?). If published, this will include your full peer review and any attached files.

Reviewer #1: No

Reviewer #2: No

---

## [Author Response · Author response to Decision Letter 0]

3 Jan 2022

Journal Requirements:

Response to Editor: We ensure that our manuscript meets PLOS ONE's style requirements.

"I have read the journal's policy and the authors of this manuscript have the following competing interests: 

Simona Granato, Giuseppe Novelli, Roberto Di Virgilio and Daria La Torre are employees of Pfizer.

Barbara Polistena declares to have received in the last 5 years payments or honoraria for lectures, presentations, speakers bureaus, manuscript writing or educational events from Allergan, Amgen, Astellas, Baxter, BMS, Boehringer-Ingelheim, Celgene, Eli Lilly, Janssen Cilag, Jazzpharma, Mylan, Nestlé HS, Novartis, Novo Nordisk, Pfizer, Roche, Sanofi, Servier, Shire, Takeda, Teva; in addition, she received consulting fees from UCB.

Federico Spandonaro declares to have received in the last 5 years payments or honoraria for lectures, presentations, speakers bureaus, manuscript writing or educational events from Allergan, Amgen, Astellas, Baxter, BMS, Boehringer-Ingelheim, Celgene, Eli Lilly, Janssen Cilag, Jazzpharma, Mylan, Nestlé HS, Novartis, Novo Nordisk, Pfizer, Roche, Sanofi, Servier, Shire, Takeda, Teva; in addition, he received consulting fees from Amgen.

All other authors declare that they have no competing interests."

We note that you received funding from a commercial source: "Allergan, Amgen, Astellas, Baxter, BMS, Boehringer-Ingelheim, Celgene, Eli Lilly, Janssen Cilag, Jazzpharma, Mylan, Nestlé HS, Novartis, Novo Nordisk, Pfizer, Roche, Sanofi, Servier, Shire, Takeda, Teva, and UCB."

Please include your amended Competing Interests Statement within your cover letter. We will change the online submission form on your behalf."

Response to Editor: We have included in the cover letter an amended Competing Interests Statements.

Response to Editor: We added at the end of the manuscript the captions of supporting information files.

Additional Editor Comments:

needs major revision in order to be considered for publication

Reviewers' comments:

Reviewer's Responses to Questions

Comments to the Author

1. Is the manuscript technically sound, and do the data support the conclusions?

Reviewer #1: Yes

Reviewer #2: Yes

2. Has the statistical analysis been performed appropriately and rigorously?

Reviewer #1: N/A

Reviewer #2: Yes

3. Have the authors made all data underlying the findings in their manuscript fully available?

Reviewer #1: Yes

Reviewer #2: Yes

4. Is the manuscript presented in an intelligible fashion and written in standard English?

Reviewer #1: Yes

Reviewer #2: Yes

5. Review Comments to the Author

Reviewer #1: The manuscript is a systematic review of the epidemiology of disorders requiring growth hormone supplementation and specific other growth hormone treatment related issues. The authors focused on studies including Italian national data. The overview is well presented with detailed methodology and informative tables containing results. The discussion is mostly summarizing the results with a few major conclusions. These are particularly exposing the possibilities to decreasing costs of GH therapy, which is of public importance in light of very high treatment costs. In general, I have no major comments regarding the manuscript. Despite the fact that part of the authors are employed in pharmaceutical companies producing GH therapeutics, I did not notice any biased reporting.

Response to Reviewer #1: We would like to thank the Reviewer #1 for his positive comments.

Reviewer #2: Summary: This is a systematic review that addresses 4 major questions regarding GH use in children in Italy: the prevalence and incidence of pediatric conditions approved as indication for GH treatment: GHD, TS, CKF, PWS, SGA, SHOX gene deficiency, adherence to GH treatment, the economic impact of GH treatment, and the quality of life of patients treated with GH and their caregivers.

Response to Reviewer #2: We would like to thank the Reviewer #2 for his comments, useful to improve the quality of our manuscript. 

Major comments:

1. Abstract, Results section: In the paragraph discussing prevalence, state the population, i.e., general population, referred patients, etc.

Response to Reviewer #2: We have specified that the prevalence of GHD and SHOX-D is in the general population of children; for Turner syndrome and PWS it is a birth prevalence, i.e. number of cases per live births.

2. Page 7, Data extraction: Clarify that there was no overlap between primary and secondary studies.

Response to Reviewer #2: We clarified this by adding the following sentence in the text: “None of the primary studies identified in this review are in common with those included in the secondary studies”.

3. Page 13, 2nd paragraph: Add a discussion about the different definitions of SGA used in the studies.

Response to Reviewer #2: We added a description of the different SGA definitions among the three studies: “This hypothetical prevalence is lower than the prevalence reported in the three population studies included in the review: 3.1% (Finland) [41] (SGA was defined as – 2 SDS of birth weight compared to a reference population), 5.5% (Sweden) [42], and 3.5% (Japan) [40] (in the last two studies, SGA was defined as − 2 SDS of birth weight or birth length compared to a reference population)”.

4a. Page 14, 1st summary bullet: Does the general population mentioned in the prevalence of GHD include both children and adults? 

Response to Reviewer #2: Thank you for giving us the chance to clarify this point. Although in the article by Tornese et al. (Medico e Bambino 2019;38:355-364) this has not been clearly specified, we have assumed that the prevalence of GHD refers to the pediatric population, owing that the whole article is about SGA children. We amended the text accordingly. 

4b. There is a striking difference between the prevalence noted in the international studies and that in Italy. (see below)

Response to Reviewer #2: We agree that the difference between the international estimates and those deriving from the Piedmont Region (Italy) registry of patients treated with GH is notably; however, should be considered that the last one (registry estimate) is an indirect estimate of GHD prevalence. The Authors of this study (Migliaretti 2006 et al., J. Endocrinol. Invest. 29: 438-442, 2006) in the Discussion section state that their prevalence estimates are higher of those reported in previous studies, probably due to different patient’s selection criteria, the cut-off used for tests allowing the diagnosis of partial or complete GHD, and to the improvements of diagnostic techniques (higher sensitivity).

We added in the discussion a text explaining this (see Response to comment n. 8).

5. The majority of the epidemiological studies are international and most have only 1 study in Italy resulting in a wide prevalence range.

Response to Reviewer #2: We pointed out in the Discussion that most of estimates come from international studies and the scarcity of Italian study. About SHOX-D, the review by Cicognani et al. states “The reason for this wide range of variability of the prevalence of SHOX-D in idiopathic short stature (ISS) is debated. Different factors may play an important role such as the various sizes of the samples, the different selection criteria, the ethnicity of the population, and the different methods used for the genetic analysis”. 

We added this sentence in the discussion: “This wide range could be due to different sample size of included studies, patient’s selection criteria, ethnicity, and different methods used for the genetic analysis”.

6. Page 20, 1st paragraph: Include the various definitions of non-adherence in the text.

Response to Reviewer #2: As suggested, we have described the non-adherence definitions for each study in the main text.

7. Page 22, 2nd paragraph: Include the reasons for discontinuation in the paper by Marciano. Was this by choice or due to prescription/supply issues?

Response to Reviewer #2: This study calculated the persistence to GH treatment based on administrative databases of drug prescriptions; therefore, the reasons for discontinuation can only be speculated. In the Discussion, Authors provided possible reasons for discontinuations. We added in the text the following description: 

“Most of those who discontinued were intermittent users (39.3%), that is, they restarted GH therapy after at least 60 days of interruption; in their paper, Marcianò et al. [23] have speculated that treatment discontinuation could be due to patients’ reduced compliance or lack of perceived benefit from GH therapy. Another possible reason for discontinuing GH therapy may be the occurrence of adverse reactions; in fact, more than 25% of patients who discontinued therapy had a new diagnosis of diabetes or cancer or has been hospitalized at least once after discontinuation. Another reason for discontinuation could be reaching final height (7.9% discontinued at age 15–17). The remainder of the subjects who discontinued therapy (27.7%) were thought to be probably related to the lack or loss of GH efficacy or to patients' lack of compliance”.

8. Discussion: Explain the differences between international and Italian estimates of GHD and PWS prevalence.

Response to Reviewer #2: About GHD, we added in the text the following sentence, according to the response to the comment n. 4b: “It should be noted that the GHD prevalence estimated by the Piedmont GH registry is higher than the prevalence reported in the literature; however, it should be considered as an indirect estimate which may differ from the real prevalence in the general population. The authors of the original study from which this estimate was reported [43], in the discussion state that their prevalence estimates are higher of those reported in previous studies, probably due to different patient selection criteria, the cut-off used for tests allowing the diagnosis of partial or complete GHD, and the improvements of diagnostic techniques (higher sensitivity)”.

Regarding PWS, the international estimates are only apparently diverse from those from Italy. In fact, the PWS prevalence of 1/15,000 (about 7 per 100,000) reported in the literature is a birth prevalence, while the reported annual incidence in Italy is referred to the entire Italian population. We added the following text in the Discussion in order to clarify this concept: “Regarding PWS, the international estimates are only apparently different from the Italian ones. In fact, the PWS prevalence of 1/15,000 (about 7 per 100,000) reported in the literature is a birth prevalence, while the reported annual incidence in Italy refers to the entire Italian population (which was on average 60.2 million per year in the period 2012-2014). Estimating the birth prevalence from the annual incidence on the total population, it would be: 37 new PWS cases/year (annual average of PWS patients in the age group 0-10 years) out of 527,845 live births (annual average of live births over the years 2012-2014), that is 7 per 100,000, exactly the same birth prevalence reported in the literature”.

9. Page 31: the quality of life assessment in patients with PWS does not seem to be specifically regarding GH treatment. Please clarify.

Response to Reviewer #2: Even if the two papers on PWS have the main focus on the disease rather than on the GH treatment, we assumed that most of included patients were treated. 

We added the following text in the quality of life paragraph: “The two studies by Lopez-Bastida [21] and Ragusa [27] focus mainly on the disease (PWS) rather than on the GH treatment. We decided to include them in our analysis because we were interested in analyzing the quality of life in patients with PWS, which is one of the diseases for which GH treatment is approved in Italy, and we assumed that most patients were treated. In the Lopez-Bastida paper [21], information on medications used by PWS patients was obtained from questionnaires and their costs calculated. In the Ragusa study [27] there were references to GH treatment in interviews with various patients and caregivers”.

Minor comments:

1. Page 34, last paragraph: Expand “CAD.”

Response to Reviewer #2: Thank you, we expanded it (Canadian Dollar).

6. PLOS authors have the option to publish the peer review history of their article (what does this mean?). If published, this will include your full peer review and any attached files.

Do you want your identity to be public for this peer review? For information about this choice, including consent withdrawal, please see our Privacy Policy.

Reviewer #1: No

Reviewer #2: No

---

## [Editor Report · Decision Letter 1]

10 Feb 2022

Pediatric growth hormone treatment in Italy: a systematic review of epidemiology, quality of life, treatment adherence, and economic impact

PONE-D-21-34417R1

Dear Dr. Orso,

We’re pleased to inform you that your manuscript has been judged scientifically suitable for publication and will be formally accepted for publication once it meets all outstanding technical requirements.

Kind regards,

Mohamed A Yassin, MD

Academic Editor

PLOS ONE

Additional Editor Comments (optional): the manuscript can be accepted in its current forms
---

## [Editor Report · Acceptance letter]

15 Feb 2022

PONE-D-21-34417R1 

Pediatric growth hormone treatment in Italy: a systematic review of epidemiology, quality of life, treatment adherence, and economic impact 

Dear Dr. Orso:

I'm pleased to inform you that your manuscript has been deemed suitable for publication in PLOS ONE. Congratulations! Your manuscript is now with our production department. 

Kind regards, 

on behalf of

Dr. Mohamed A Yassin 

Academic Editor

PLOS ONE